# LONGRLVR:
# LONG-CONTEXT REINFORCEMENT LEARNING REQUIRES VERIFIABLE CONTEXT REWARDS

**Guanzheng Chen**[1,2,3*]    **Michael Qizhe Shieh**[1,3†]    **Lidong Bing**[2]
[1]National University of Singapore    [2]MiroMind AI    [3]absolute AI
gc.chen@u.nus.edu, michaelshieh@comp.nus.edu.sg, lidong.bing@shanda.com

## ABSTRACT

Reinforcement Learning with Verifiable Rewards (RLVR) has significantly advanced the reasoning capabilities of Large Language Models (LLMs) by optimizing them against factual outcomes. However, this paradigm falters in long-context scenarios, as its reliance on *internal* parametric knowledge is ill-suited for tasks requiring *contextual grounding*—the ability to find and reason over *externally* provided information. We identify a key reason for this failure: a reward based solely on the final answer is too sparse to effectively guide the model for identifying relevant evidence. We formally prove that the outcome-only reward leads to significant vanishing gradients for the context grounding process, rendering learning intractable. To overcome this bottleneck, we introduce **LongRLVR** to augment the sparse answer reward with a dense and *verifiable context reward*. This auxiliary signal directly incentivizes the model for selecting the correct grounding information, providing a robust learning gradient that solves the underlying optimization challenge. We validate our method on challenging long-context benchmarks using Qwen and LLaMA models. LongRLVR consistently and significantly outperforms the standard RLVR across all models and benchmarks, e.g., boosting a 14B model's scores on RULER-QA from 73.17 to 88.90 and on LongBench v2 from 39.8 to 46.5. Our work demonstrates that explicitly rewarding the grounding process is a critical and effective strategy for unlocking the full reasoning potential of LLMs in long-context applications. Our code is available at `https://github.com/real-absolute-AI/LongRLVR`.

## 1 INTRODUCTION

Reinforcement Learning with Verifiable Rewards (RLVR) (Lambert et al., 2024; Guo et al., 2025) has emerged as a pivotal paradigm in advancing the reasoning capabilities of Large Language Models (LLMs). By rewarding verifiable outcomes, RLVR effectively steers LLMs to explore diverse reasoning pathways for achieving factually accurate and logically sound solutions. This paradigm has recently propelled LLMs, such as DeepSeek-R1 (Guo et al., 2025), to achieve expert-level reasoning ability in domains like mathematics and programming (Guo et al., 2025; Jaech et al., 2024; Kimi et al., 2025; Huang & Yang, 2025). The remarkable success of RLVR on complex reasoning makes it never more compelling for applying to the next frontier: enabling LLMs to explore and reason over vast external environment to unlock broader intelligence (Zhang et al., 2025). However, the interaction of LLMs with such environments necessitates processing extensive external information, which poses significant challenges on their long-context capabilities.

Effective long-context reasoning typically hinges upon robust contextual grounding: the ability to accurately retrieve and synthesize information from *external* documents (Wan et al., 2025). Yet, recent studies (Yue et al., 2025; Wen et al., 2025) suggest that RLVR primarily sharpens the *internal* knowledge that LLMs have already acquired during pretraining. This may limit the efficacy of RLVR for enhancing the long-context capabilities of LLMs.

---

*Work done during the internship of Guanzheng Chen at MiroMind AI.
†Corresponding Author.

As shown in Figure 1, when applying naive RLVR with outcome-only rewards for final answers upon long-context training, the model's contextual recall score (measured by retrieving reference chunks identifiers as detailed in Figure 2) quickly stagnates. This plateau in relevant retrieval directly creates a ceiling for answer accuracy, thus halting overall learning progress on training rewards.

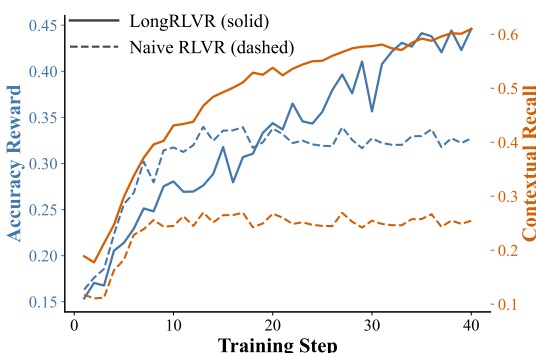

Figure 1: The accuracy reward and contextual recall of naive RLVR and LongRLVR on the training data.

In this work, we introduce LongRLVR to address the bottleneck of naive RLVR on long-context training. We first formally prove that the outcome-only reward causes vanishing gradients for the contextual grounding, rendering the learning to become sparse and intractable for long sequences. To address this, LongRLVR incorporates a *context reward* into outcome-only rewards to augment the sparse learning signal on contextual grounding. Specifically, for each rollout, we steer the model to generate the grounding chunk identifiers from the long context before achieving the final answer (see Figure 2). These identifiers will be compared with ground-truth counterparts to access a verifiable reward. By explicitly rewarding the model for extracting relevant evidences, we provide a dense learning signal that mitigates the vanishing gradient issue. Therefore, our LongRLVR overcomes the bottleneck of long-context RLVR training and allows both contextual recall and answer accuracy to improve continuously throughout training (see Figure 1).

To support the training of LongRLVR, we develop a comprehensive data synthetic pipeline that produces high-quality, long-context question-answering data annotated with the necessary grounding chunks. We validate its effectiveness through extensive experiments on LLaMA-3.1 (Dubey et al., 2024) and Qwen2.5 (Yang et al., 2025) models across challenging long-context benchmarks such as RULER (Hsieh et al., 2024), LongBench v2 (Bai et al., 2024b), and LongReason (Ling et al., 2025). Our method consistently and significantly outperforms the outcome-only RLVR baseline. For instance, LongRLVR largely catapults the score of Qwen2.5-14B-1M across all benchmarks ($73.17 \rightarrow 88.90$ on RULER-QA, $40.2 \rightarrow 46.5$ on LongBench v2, and $73.55 \rightarrow 78.42$ on LongReason). By successfully training models to ground their reasoning in provided context, LongRLVR not only overcomes the limitations of conventional RLVR but empower these models with remarkable long-context reasoning abilities comparable with, and even superior to, state-of-the-art reasoning models such as Qwen3 (Qwen, 2025) series.

## 2 METHOD

In this section, we introduce LongRLVR to remedy the limitations of RLVR in long-context tasks. We first present an explicit grounding formulation for long-context RLVR in §2.1. Next, in §2.2, we formally prove that outcome-only rewards lead to a vanishing gradient problem for this grounding process. To solve this, we introduce our verifiable context reward, presenting its theoretical foundation in §2.3.1 and a practical F-score-based implementation in §2.3.2. Finally, we detail the synthetic data generation pipeline that enables this approach in §2.4.

### 2.1 RLVR ON LONG CONTEXTS: AN EXPLICIT GROUNDING FORMULATION

The standard RLVR framework aims to optimize a policy $\pi_\theta(y \mid X, Q)$ that generates an answer $y$ given a context $X$ and a question $Q$. The objective is to maximize the expected verifiable reward $r_{\text{ans}}(y)$, which typically evaluates the correctness of the final answer:

$$J_{\text{ans}}(\theta) = \mathbb{E}_{(X,Q)\sim\mathcal{D}} \left[ \mathbb{E}_{y\sim\pi_\theta(y|X,Q)}[r_{\text{ans}}(y)] \right]. \tag{1}$$

This formulation, while effective for tasks where reasoning relies on parametric knowledge, ignores two distinct processes in long-context scenarios: (1) **contextual grounding**, the act of identifying the relevant subset of information within $X$, and (2) **answer generation**, the act of synthesizing an answer from the grounded information. When the context $X$ is extensive, the grounding process becomes non-trivial yet remains implicit within the monolithic policy $\pi_\theta(y \mid X, Q)$.

Here, we refactor the policy to explicitly model these two stages. Let the long context $X$ be segmented into a set of $N$ chunks, $C = \{c_1, \ldots, c_N\}$, the long-context policy should jointly involve grounding and answering to identify a subset of selected chunks $Z \subseteq C$ and a final answer $y$. This process is modeled as a factorized distribution:

$$\pi_\theta(y, Z \mid X, Q) = \underbrace{\pi_\theta^{\text{gnd}}(Z \mid X, Q)}_{\text{Grounding Head}} \cdot \underbrace{\pi_\theta^{\text{ans}}(y \mid X, Q, Z)}_{\text{Answer Head}}. \tag{2}$$

The **Grounding Head** is responsible for contextual grounding, selecting the evidence $Z$ required to answer the question. The **Answer Head** then conditions on this selected evidence to produce the final output $y$.

## 2.2 THE VANISHING GROUNDING GRADIENT WITH OUTCOME-ONLY REWARDS

We now formally analyze the learning dynamics of the factorized policy (Eq. 2) when optimized solely with the final answer reward, $r_{\text{ans}}(y)$. We will demonstrate that this outcome-only signal is insufficient for learning the grounding head ($\pi_\theta^{\text{gnd}}$), creating a fundamental bottleneck for long-context reasoning.

Our analysis is based on a common property of long-context reasoning tasks: a correct solution often requires synthesizing a complete set of prerequisite evidence. Partial information, while helpful, typically yields a lower reward. That said, an LLM may occasionally answer correctly from a subset of $G$ or from alternative supporting evidence. This structure motivates the following formal assumption.

**Assumption 1** (Sparse Answer Reward). *Let $G \subseteq C$ be the ground-truth set of essential evidence chunks. There exists a non-negative, monotone set function $f : 2^G \to \mathbb{R}_{\geq 0}$ with $f(\emptyset) = 0$ such that the expected answer reward conditioned on the selected set $Z$ depends only on which ground-truth chunks are present:*

$$\mathbb{E}[r_{ans} \mid Z] = \mu_0 + f(Z \cap G), \tag{3}$$

*where $\mu_0$ is a baseline reward from partial or spurious evidence. This form allows different chunks in $G$ to have different importance and credits arbitrary subsets $Z \cap G$.*

To analyze the gradient, we introduce a logit $s_j$ for each chunk $c_j \in C$ and denote by $z_j = \mathbf{1}\{c_j \in Z\}$ its selection indicator. Let $p_j = \Pr_\theta(c_j \in Z) = \mathbb{E}_\theta[z_j]$ be the marginal selection probability under the grounding policy, we can derive the proposition below.

**Proposition 1** (Vanishing Gradients for Grounding). *Under Assumption 1 and the grounding parameterization in Eq. (9), the gradient of the expected answer reward with respect to the logit $s_j$ for any essential chunk $c_j \in G$ is:*

$$\nabla_{s_j} \mathbb{E}[r_{ans}] = \text{Cov}\big(f(Z \cap G), z_j\big) = p_j(1 - p_j)\big(\mathbb{E}[f(Z \cap G) \mid z_j{=}1] - \mathbb{E}[f(Z \cap G) \mid z_j{=}0]\big).$$

*Let $\Delta_j(T) \triangleq f(T \cup \{c_j\}) - f(T)$ denote the marginal gain of chunk $c_j$ for any $T \subseteq G \setminus \{c_j\}$, and assume $\Delta_j(T) \leq \bar{\delta}_j$ for some constant $\bar{\delta}_j > 0$. Define the* activation event *for $c_j$*

$$\mathcal{E}_j \triangleq \big\{Z : \Delta_j\big((Z \cap G) \setminus \{c_j\}\big) > 0\big\},$$

*i.e., the event that the rest of the prerequisite evidence that makes $c_j$ useful is already present in $Z$. Then*

$$0 \leq \nabla_{s_j} \mathbb{E}[r_{ans}] \leq p_j(1 - p_j)\, \bar{\delta}_j \Pr_\theta(\mathcal{E}_j).$$

*(See proof in Appx. §A.2.)*

Proposition 1 shows that the learning signal for selecting any single required chunk $c_j$ is scaled by $\Pr_\theta(\mathcal{E}_j)$—the probability that *all of the other prerequisite evidence that interacts with $c_j$ has already been selected*. In challenging long-context tasks where correctly answering the question requires combining many pieces of *implicit* evidence, this activation event is extremely unlikely under the initial RLVR policy: a single rollout must simultaneously include a large subset of $G$ before $c_j$ can receive positive credit. Consequently, the answer-only gradient for $c_j$ is suppressed by the tiny factor $\Pr_\theta(\mathcal{E}_j)$ and becomes effectively zero for many ground-truth chunks early in training. Once these gradients vanish due to small standard deviation of context rewards (Razin et al., 2023), the grounding head is non-trivial to increase the selection probability of the corresponding evidence, causing contextual recall to stagnate and inducing the plateau in training reward observed in Figure 1.

## 2.3 LongRLVR: Learning with a Verifiable Context Reward

To surmount the vanishing gradient problem introduced in §2.2, we propose augmenting the sparse, outcome-only reward with a direct, dense signal that supervises the grounding head. The core is the incorporation of a **verifiable context reward**, $r_{\text{ctx}}$, which provides a granular learning signal for the contextual grounding process.

### 2.3.1 Theoretical Foundation

We begin by defining a general class of context rewards as any function that increases whenever an additional ground-truth chunk in $G$ is correctly selected, i.e., a reward that is monotone in the *set* $Z \cap G$ rather than only in its cardinality. Different chunks may contribute different amounts. For analytical tractability, we consider a simple additive form that assigns a (possibly distinct) weight to each ground-truth chunk:

$$r_{\text{ctx}}(Z, G) = \sum_{c_k \in G} \alpha_k \mathbf{1}\{c_k \in Z\}, \tag{4}$$

where $\alpha_k > 0$ controls the contribution of chunk $c_k$. This formulation ensures the policy receives positive feedback for each relevant chunk it selects, irrespective of whether the complete evidence set $G$ is recovered.

The final reward in the LongRLVR framework is a linear combination of the answer and context rewards:

$$r_{\text{total}}(y, Z) = r_{\text{ans}}(y) + r_{\text{ctx}}(Z, G). \tag{5}$$

We then prove this general structure is sufficient to provably resolve the vanishing gradient problem.

**Proposition 2** (Non-Vanishing Grounding Signal). *For the total reward $r_{total} = r_{ans} + r_{ctx}$ with $r_{ctx}(Z, G) = \sum_{c_k \in G} \alpha_k \mathbf{1}\{c_k \in Z\}$, the gradient of the expected total reward with respect to the logit $s_j$ for any essential chunk $c_j \in G$ is (see proof in Appx. §A.3)*

$$\nabla_{s_j} \mathbb{E}[r_{total}] = \underbrace{\nabla_{s_j} \mathbb{E}[r_{ans}]}_{\text{From } r_{ans}} + \underbrace{\alpha_j \text{Var}(z_j) + \sum_{\substack{k \neq j \\ c_k \in G}} \alpha_k \text{Cov}(z_k, z_j)}_{\text{From } r_{ctx}}.$$

*In particular, combining this with Proposition 1 shows that the answer-only term is at most $p_j(1 - p_j) \bar{\delta}_j \Pr_\theta(\mathcal{E}_j)$, while the context term always contains the* dense *component $\alpha_j \text{Var}(z_j) = \alpha_j p_j(1 - p_j)$ that is not multiplied by $\Pr_\theta(\mathcal{E}_j)$. If the grounding policy tends to select related chunks together (so that $\text{Cov}(z_k, z_j) \geq 0$ for $k \neq j$), the cross-covariance terms further strengthen this signal.*

The second term in Proposition 2 thus provides a dense learning signal for each chunk that is independent of the rare activation event $\mathcal{E}_j$, preventing the gradient from vanishing even when the answer-only component is negligible. This theoretical foundation establishes that rewarding intermediate grounding steps—at the level of actual chunks rather than just outcome correctness—is a sound and effective strategy for overcoming the learning bottleneck in long-context RLVR.

The second term provides a dense learning signal for each chunk that is independent of the joint success probability $q$, preventing the gradient from vanishing. This theoretical foundation establishes that rewarding intermediate grounding steps is a sound and effective strategy for overcoming the learning bottleneck in long-context RLVR.

### 2.3.2 A Practical Instantiation: The Modulated F-Score Reward

While our general formulation guarantees a non-vanishing gradient, a well-designed, normalized reward is crucial for stable and effective training. A naive metric like recall ($|Z \cap G|/|G|$) is insufficient, as it would incentivize a degenerate policy of selecting all available chunks. A practical reward must balance the retrieval of correct evidence (recall) with the avoidance of irrelevant information (precision).

```
<long_context>
<CHUNK_1> Marie Curie was born in Warsaw, Poland... she moved to Paris to
pursue higher education... </CHUNK_1>
<CHUNK_2> The Curies' early research was inspired by Henri Becquerel's 1896
discovery... </CHUNK_2>
...
<CHUNK_N> In December 1898, they announced the discovery of a second ele
ment, "radium,"... </CHUNK_N>
</long_context>
- - - - - - - - - - - - - - - - - - - - - - - - - - - - - - - - - - - - - - - -
Question: Where was Marie Curie born and what was the second radioactive
element she co-discovered?
- - - - - - - - - - - - - - - - - - - - - - - - - - - - - - - - - - - - - - - -

Output:
<think> ... </think>

<useful_chunks> <CHUNK_1>, <CHUNK_N> </useful_chunks>

<answer> Marie Curie was born in Warsaw, Poland, and the second
radioactive element she co-discovered was radium. </answer>
```

Figure 2: Data format for LongRLVR training. The model is tasked to retrieve useful chunks from the long context before generating the final answer. These chunk identifiers are utilized to derive verifiable context rewards.

To this end, we adopt the $F_\beta$-score as the core measure of grounding quality. The $F_\beta$-score is the weighted harmonic mean of precision and recall:

$$F_\beta(Z, G) = (1 + \beta^2) \frac{\text{Precision}(Z, G) \cdot \text{Recall}(Z, G)}{(\beta^2 \cdot \text{Precision}(Z, G)) + \text{Recall}(Z, G)}, \tag{6}$$

where $\beta$ is a parameter that allows us to weigh recall more heavily than precision (e.g., $\beta = 2$), ensuring the model is primarily incentivized to gather all necessary evidence.

To create a synergistic effect between grounding and final answer accuracy, we formulate our context reward as a modulated combination of the $F_\beta$-score and the answer reward:

$$r_{\text{ctx}}(y, Z, G) = \eta \cdot F_\beta(Z, G) + (1 - \eta) \cdot r_{\text{ans}}(y) \cdot F_\beta(Z, G), \tag{7}$$

where $\eta \in [0, 1]$ is a blending hyperparameter. This reward structure has two key components: (1) **Unconditional Grounding Reward** ($\eta \cdot F_\beta$): This term provides a dense, stable reward for selecting correct evidence, ensuring the grounding head always receives a learning signal. (2) **Synergistic Success Reward** ($(1 - \eta) \cdot r_{\text{ans}} \cdot F_\beta$): This component acts as a synergistic gate, ensuring that the full reward for high-quality grounding is unlocked only upon generating a correct answer. It incentivizes the model to treat accurate grounding as a means to a correct final answer, unifying both objectives and preventing the policy from perfecting grounding in isolation.

With our proposed context reward, the final LongRLVR objective is to maximize the expected total reward over the data distribution and the stochastic policy:

$$J(\theta) = \mathbb{E}_{(X,Q,G) \sim \mathcal{D}} \left[ \mathbb{E}_{(Z,y) \sim \pi_\theta(Z,y|X,Q)} \left[ r_{\text{ans}}(y) + r_{\text{ctx}}(y, Z, G) \right] \right]. \tag{8}$$

This objective can be optimized using standard policy gradient algorithms such as PPO and GRPO. To facilitate the computation of $r_{\text{ctx}}$, we design the policy to first generate a list of identifiers for the selected chunks ($Z$) before generating the final answer ($y$), as illustrated in Figure 2.

## 2.4 SYNTHETIC DATA GENERATION FOR GROUNDED QA

Training LongRLVR necessitates a specialized dataset comprising tuples of $(X, Q, G, y)$, where $G$ is the ground-truth set of evidence chunks from context $X$ essential for answering question $Q$ with

answer $y$. As such datasets are exceedingly rare, we developed the automated pipeline detailed in Algorithm 1 to produce high-fidelity, challenging QA pairs with precise grounding annotations. This pipeline is crucial for the direct supervision of the contextual grounding mechanism in our model.

---

**Algorithm 1** Synthetic Data Generation Pipeline for Grounded QA

---

1: **Input:** A collection of long documents $\mathcal{X}$.
2: **Output:** A filtered dataset $\mathcal{D} = \{(X, Q, G, y)\}$.
3: **for** each document $X \in \mathcal{X}$ **do**
4:     **// Step 1: Semantic Clustering and Evidence Identification**
5:     Partition $X$ into a set of text chunks $C = \{c_1, \ldots, c_N\}$.
6:     Embed all chunks into a dense vector space using a sentence encoder.
7:     Apply a density-based clustering algorithm to the embeddings to form thematic clusters $\mathcal{K} = \{K_1, K_2, \ldots\}$.
8:     **// Step 2: Per-Cluster QA Generation and Scoring**
9:     Initialize a set of best-per-cluster candidates, $\mathcal{S}_{\text{doc}} \leftarrow \emptyset$.
10:     **for** each cluster $K_i \in \mathcal{K}$ **do**
11:         **Generate Candidates:** Prompt a generator LLM with the content of $K_i$ to synthesize $k$ candidate tuples $\{(Q_j, y_j, G_j)\}_{j=1}^{k}$.
12:             ▷ Crucially, the LLM itself identifies the necessary evidence $G_j \subseteq K_i$ for each QA pair.
13:         **Score Candidates:** For each candidate tuple, use a verifier LLM to assign a quality score $s_j$ based on question clarity, answer fidelity, and evidence necessity.
14:         **Intra-Cluster Selection (Stage 1):** Identify the candidate $(Q_i^*, y_i^*, G_i^*)$ with the highest score $s_i^*$ within the cluster.
15:         Add the highest-scoring tuple $(Q_i^*, y_i^*, G_i^*, s_i^*)$ to $\mathcal{S}_{\text{doc}}$.
16:     **// Step 3: Inter-Cluster Selection and Finalization (Stage 2)**
17:     Select the tuple $(Q^*, y^*, G^*)$ from $\mathcal{S}_{\text{doc}}$ that has the overall highest score, breaking ties randomly.
18:     Add the final, document-best tuple $(X, Q^*, G^*, y^*)$ to the dataset $\mathcal{D}$.
19: **return** $\mathcal{D}$

---

This automated, multi-stage pipeline enables the scalable creation of challenging long-context QA examples with the explicit evidence annotations required to compute our verifiable context reward.

## 3 Experimental Setup

### 3.1 Implementation Details

**Data Curation.** To train our model, we constructed a large-scale, high-quality dataset of 46K long-context question-answering pairs with explicit grounding annotations. We sourced documents from book, arXiv, and code domains, filtering for lengths between 8K and 64K tokens. Following the pipeline detailed in Alg. 1, we first identified semantically coherent clusters of text segments within each document. For each document, we then used a powerful generator model, Qwen3-235B-A22B (Qwen, 2025), to create multiple candidate QA pairs, with each answer grounded in specific evidence segments. To ensure the highest quality, the same model was used as a judge to score the correctness and evidence relevance of each pair. A two-stage rejection sampling process selected the single best QA pair per document, and we applied a strict final filter, retaining only pairs with a quality rating above 9 out of 10. See more details in Appx. §B.

**Training Details.** We train three models: LLaMA-3.1-8B, Qwen2.5-7B-1M, and Qwen2.5-14B-1M [1] with RLVR implemented by naive Group Relative Policy Optimization (GRPO) (Shao et al., 2024). Crucially, before training of each model, we exclude easy questions for which its answer upon full long context is rated 8 or higher by a Qwen3-A235B-A22B judge. For the RL training, we use the AdamW optimizer with a constant learning rate of 1e-6 and a 5-step linear warmup. During rollouts, we use a prompt batch size of 512 and sample 8 responses per prompt, with a maximum context length of 64K and a response length of 4096. We train all models for one epoch on 46K crafted data. For hyperparameters, we set $\eta$ as 0.1 and $\beta$ as 2 in Eq. (7).

---

[1] All models refer to the instruct version.

Table 1: The evaluation of models on long-context benchmarks. The metric in all benchmarks is accuracy. The best score across all models is highlighted in **green**, and the second-best is in **red**. Additionally, the best score within each trained model comparing among SFT, RLVR, and our LongRLVR is bolded.

| Model | RULER-QA | | | | LongBench v2 | | | | LongReason | | | |
|---|---|---|---|---|---|---|---|---|---|---|---|---|
| | 32K | 64K | 128K | AVG | Short | Medium | Long | Overall | 32k | 64k | 128k | Avg. |
| LLaMA-3.1-70B | 70.4 | 64.2 | 47.6 | 60.73 | 36.2 | **45.0** | 34.0 | 25.9 | 61.16 | 63.30 | 48.30 | 57.59 |
| Qwen2.5-72B-YaRN | 66.9 | 54.5 | 47.2 | 56.20 | 43.5 | **48.9** | 40.9 | 43.5 | 74.27 | 74.53 | 69.48 | 72.76 |
| Qwen3-8B (Thinking) | 86.5 | 84.0 | 81.8 | 84.10 | 43.3 | 28.8 | 32.4 | 37.6 | 77.23 | 71.28 | 65.99 | 71.50 |
| Qwen3-14B (Thinking) | **91.2** | **89.0** | **82.6** | **87.60** | 51.7 | 42.3 | **38.9** | **44.9** | 80.86 | 77.08 | 74.56 | 77.50 |
| QwenLong-L1-32B | 89.0 | 77.0 | 72.4 | 79.47 | **53.3** | 34.4 | 33.3 | 41.0 | **84.13** | **83.63** | 75.06 | **80.94** |
| LLaMA-3.1-8B | 65.8 | 63.7 | 58.8 | 62.77 | 34.4 | **31.6** | 21.3 | 30.4 | 51.45 | 49.94 | 46.53 | 49.31 |
| -SFT | 68.4 | 65.3 | 60.4 | 64.70 | 36.1 | 28.4 | 28.7 | 31.2 | 50.88 | 49.11 | 48.87 | 49.62 |
| -RLVR | 72.0 | 68.8 | 62.6 | 67.80 | 35.6 | 31.2 | 29.6 | 32.4 | 49.87 | 49.62 | 49.37 | 49.62 |
| -LongRLVR | **85.5** | **76.5** | **79.0** | **80.33** | **41.1** | 30.7 | **38.9** | **36.2** | **51.89** | **51.01** | **56.80** | **53.23** |
| Qwen2.5-7B-1M | 70.5 | 66.0 | 58.5 | 65.00 | 37.8 | 31.2 | 28.7 | 33.0 | 66.75 | 66.25 | 66.36 | 66.45 |
| -SFT | 72.4 | 64.2 | 56.8 | 64.47 | 36.7 | 32.6 | 28.7 | 33.2 | 68.64 | 66.83 | 66.62 | 67.36 |
| -RLVR | 74.4 | 68.5 | 57.8 | 66.90 | 37.2 | 29.3 | 30.6 | 32.4 | 70.78 | 69.02 | 68.01 | 69.27 |
| -LongRLVR | **82.5** | **76.5** | **77.0** | **78.67** | **45.6** | **35.8** | 32.4 | **38.6** | **80.35** | **79.47** | **77.83** | **79.22** |
| Qwen2.5-14B-1M | 90.6 | 70.6 | 64.4 | 75.20 | 51.7 | 34.0 | 33.3 | 40.2 | 75.44 | 71.79 | 73.42 | 73.55 |
| -SFT | 88.0 | 66.5 | 62.2 | 72.23 | 48.9 | 34.9 | 33.3 | 39.6 | 74.18 | 70.03 | 69.27 | 71.16 |
| -RLVR | 86.3 | 69.0 | 64.2 | 73.17 | 48.3 | 36.7 | 31.5 | 39.8 | 74.06 | 71.91 | 71.03 | 72.33 |
| -LongRLVR | **95.4** | **87.8** | **83.5** | **88.90** | **55.6** | **43.3** | **38.0** | **46.5** | **81.23** | **77.96** | **76.07** | **78.42** |

## 3.2 EVALUATION PROTOCOL

**Baselines.** We compare LongRLVR against two controlled baselines: Supervised Fine-Tuning (SFT) and naive RLVR (GRPO). All methods are applied to LLaMA-3.1-8B, Qwen2.5-7B-1M, and Qwen2.5-14B-1M, using the same synthetic training data. To contextualize performance, we also report scores for leading open-source models (LLaMA-3.1-70B, Qwen2.5-72B, Qwen3 series) and a specialized long-context baseline, QwenLong-L1-32B (Wan et al., 2025). The context windows of Qwen2.5-72B and Qwen3 models are extended to 128K using YaRN (Peng et al., 2023), while Qwen3 models are evaluated in their thinking mode.

**Benchmarks.** We evaluate all models on three challenging long-context QA benchmarks: (1) **RULER-QA** (Hsieh et al., 2024): A synthetic benchmark testing multi-hop reasoning over arbitrary context length. We focus on this QA task with the lengths of 32K, 64K, and 128K. (2) **LongBench v2** (Bai et al., 2024b): A realistic multi-choice QA benchmark on documents up to 128K tokens. Standard baselines are evaluated with CoT, while models that output reasoning steps (ours and the Qwen3 series) are evaluated on their final answer. (3) **LongReason** (Ling et al., 2025): A synthetic multi-choice benchmark designed for controllable evaluation of long-context reasoning. We evaluate the lengths of 32K, 64K, and 128K.

## 4 RESULTS AND ANALYSES

### 4.1 MAIN RESULTS

In Table 1, we present the comprehensive evaluation of LongRLVR against various baselines. The results reveal the exceptional effectiveness of our approach, which we analyze through two critical comparisons: (1) against naive SFT and RLVR baselines to demonstrate consistent and substantial gains, and (2) against superior LLMs to establish its competitiveness.

**Consistent and substantial gains over naive SFT and RLVR.** LongRLVR consistently and substantially outperforms both SFT and naive RLVR when applied to the same base models with identical training data. This is established across different model families (LLaMA and Qwen) and scales (7B, 8B, and 14B), confirming the general applicability of our approach. For instance, LongRLVR achieves large gains over naive RLVR across all benchmarks and models: for Qwen2.5-14B-1M

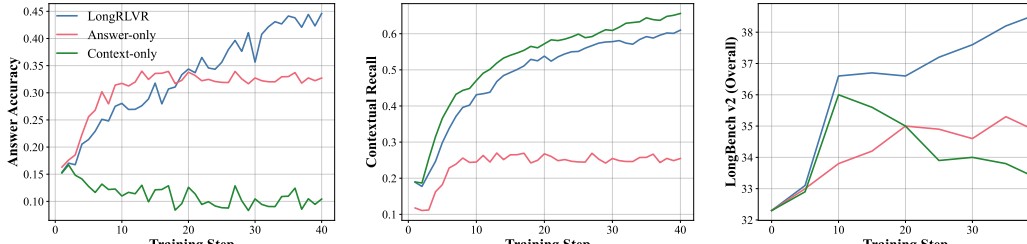

Figure 3: Study on reward components. The answer-only model suffers from stagnating contextual recall, which caps its final performance. The context-only model excels at recall but fails to achieve accurate rewards. By synergizing both signals, Qwen2.5-7B-1M-LongRLVR achieves the best and most stable performance on the LongBench v2 benchmark, proving that both rewards are essential.

(e.g., 46.5 vs. 39.8 on LongBench v2), Qwen2.5-7B-1M (e.g., 38.6 vs. 32.4 on LongBench v2), and LLaMA-3.1-8B (e.g., 36.2 vs. 32.4 on LongBench v2). The consistency of these large gains provides strong evidence that LongRLVR effectively remedies the fundamental limitations of naive RLVR on long-context scenarios by directly supervising the contextual grounding process. In addition, the superiority to SFT demonstrates the potential of RLVR as a compelling post-training approach for incentivizing long-context capabilities.

**Comparable to superior LLMs.** Beyond outperforming direct RLVR, LongRLVR elevates LLMs to a exceptional performance tier, enabling them to surpass much larger conventional models and rival the latest specialized reasoning LLMs. First, our LongRLVR demonstrates remarkable parameter efficiency against larger, conventional LLMs. Our Qwen2.5-7B-1M model (79.22 on LongReason) significantly outperforms both the LLaMA-3.1-70B (57.59) and the Qwen2.5-72B-YaRN (72.76). Similarly, our 14B model (46.5 on LongBench v2) even surpass the performance of the 72B model, showcasing the ability to instill powerful long-context reasoning capabilities in a much smaller parameter footprint. Second, LongRLVR empower conventional base models with exceptional long-context reasoning abilities that compete with and even surpass specialized models. Notably, our Qwen2.5-14B-1M, trained with LongRLVR, outperforming the newer Qwen3-14B (88.90 vs 87.60 on RULER-QA, 78.42 vs 77.50 on LongReason) which benefits from a more advanced backbone and post-training strategy. Moreover, our 14B model is comparable to the much larger QwenLong-L1-32B, which derives from the reasoning model, R1-Distilled-Qwen-32B, trained with long-context RLVR. This demonstrates the significant effectiveness our method to unlock superior long-context reasoning for non-reasoning LLMs.

## 4.2 IMPACT OF REWARD COMPONENTS

In Figure 1, we demonstrate that our LongRLVR overcomes the bottleneck of outcome-based RLVR by incorporating verifiable context rewards. To isolate the impact of the context reward, in Figure 3, we compare the training of Qwen2.5-7B-1M with the full LongRLVR against using answer-only and context-only ($F_\beta$ score in Eq. (7)) rewards, respectively. The results confirm our central hypothesis that the contextual recall of answer-only baseline quickly stagnates, thus creating a hard performance ceiling on both the training reward and the downstream task. Conversely, the model trained with context-only reward, despite involving a flat answer reward, shows rapid initial performance gains on the LongBench v2 benchmark. This demonstrates that mastering contextual grounding is a foundational capabilities that directly boosts long-context reasoning. However, without the final answer reward to steer reasoning toward a correct outcome, its downstream performance eventually degrades. While our LongRLVR succeeds by synergizing both signals, hence achieving continually improved training answer reward and downstream tasks performance.

## 4.3 IMPACT OF DATA QUALITY

We study the impact of our two data quality strategies in our data synthetic pipeline: (1) using rejection sampling to select high-quality generated QA pairs, and (2) filtering out easy questions. We ablate these choices using the Qwen2.5-7B-1M model and report the overall score on LongBench v2. The results are shown in Figure 4. First, Figure 4 (left) shows that rejection sampling quality is

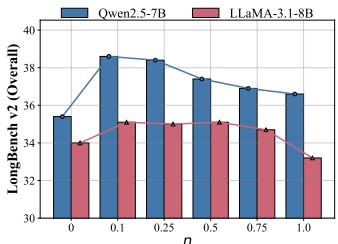 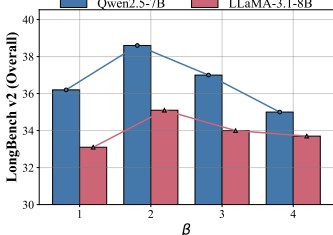 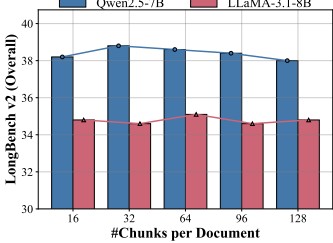

(a) Effect of blending factor $\eta$.     (b) Effect of F-score parameter $\beta$.     (c) Robustness to chunk number.

Figure 5: Ablation studies on key hyperparameters for LongRLVR. We analyze the overall performance on LongBench v2 while varying (a) the blending factor $\eta$ in the context reward, (b) the F-score parameter $\beta$, and (c) the number of chunks per document. Results are reported for both Qwen2.5-7B and LLaMA-3.1-8B.

critical. Using the best-rated samples achieves our top score of 38.6, which degrades significantly with median (36.6) and worst-rated (34.8) samples. Second, Figure 4 (right) analyzes our filtering strategy. Our default method of filtering only easy questions proves most effective. Crucially, filtering out *hard* questions is highly detrimental, causing performance to plummet to 35.8, nearly as low as applying no filtering at all (35.6). This suggests that these challenging examples are essential for enhancing the complex reasoning ability required for long-context tasks.

## 4.4 ABLATION STUDIES ON HYPERPARAMETERS

We further conduct ablation studies to analyze key hyperparameters in our method, with results shown in Figure 5. (1) **Blending Factor $\eta$.** This factor balances the unconditional grounding reward ($F_\beta$) and the synergistic reward ($r_{\text{ans}} \cdot F_\beta$). Figure 5a shows that performance peaks at a small, non-zero value ($\eta = 0.1$). A purely synergistic reward ($\eta = 0$) is suboptimal because the initial learning signal is too sparse. Conversely, a purely unconditional reward ($\eta = 1$) decouples grounding from the final goal of producing a correct answer, hence leading to inferior effectiveness. (2) **F-score Parameter $\beta$.** The $\beta$ parameter trades off recall and precision in the grounding reward. As shown in Figure 5b, performance is optimal at $\beta = 2$. This moderately prioritizes recall, which is critical for complex reasoning where failing to retrieve a single essential piece of evidence can be catastrophic. A lower $\beta$ encourages an overly conservative policy that fails to retrieve all necessary chunks, while a higher $\beta$

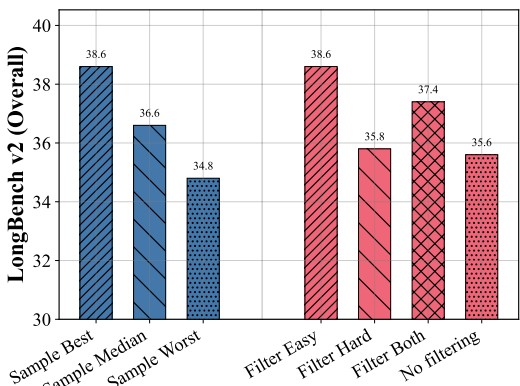

Figure 4: **Data quality ablation on LongBench v2.** Left: The effect of rejection sampling quality. Right: The effect of different data filtering strategies. High-quality, challenging data is shown to be most effective. Results are reported on Qwen2.5-7B-1M-LongRLVR.

incentivizes retrieving too much irrelevant context, which complicates the final reasoning step. (3) **Robustness to Number of Chunks.** Figure 5c demonstrates that LongRLVR is remarkably robust to the number of chunks per document, maintaining high performance from 16 to 128 chunks per document during evaluation. This is a significant practical advantage over traditional retrieval systems, which are often highly sensitive to chunking strategy. This robustness indicates that the model learns a flexible semantic grounding policy rather than relying on surface-level features, allowing it to identify relevant information regardless of how it is segmented.

## 5 RELATED WORK

**Reinforcement Learning with Verifiable Rewards (RLVR).** Reinforcement Learning with Verifiable Rewards (RLVR) has emerged as a powerful paradigm for enhancing the reasoning of LLMs by rewarding models based on deterministic, ground-truth outcomes like passing unit tests or deriving a correct solution (Lambert et al., 2024; Guo et al., 2025). This approach has propelled models to expert-level (e.g., IMO-level mathematics) performance on complex, self-contained reasoning tasks such as mathematics and programming (Guo et al., 2025; Jaech et al., 2024; Kimi et al., 2025; Huang & Yang, 2025). In these settings, the primary challenge is to refine the model's internal, parametric knowledge to discover a correct chain of thought (Yue et al., 2025; Wen et al., 2025). However, the efficacy of this outcome-only reward structure is limited in long-context scenarios, where success hinges first on identifying relevant evidence from a vast external input—a process we term contextual grounding (Wan et al., 2025). Wang et al. (2025) incorporate retrieval reward in RLVR for the appearance of correct context in thinking process. Our work directly addresses this gap by introducing a verifiable reward for the intermediate grounding process itself.

**Long Context Alignment.** Previous studies successfully extended model context windows through methods like Rotary Position Embedding (RoPE) scaling (Su et al., 2022; Chen et al., 2023; Peng et al., 2023; An et al., 2024). Yet, the extended models with long context windows often fail to reliably use the information in applications. To solve this, long-context alignment becomes crucial to unlock the model's latent capabilities by post training, which includes long-context SFT (Bai et al., 2024a), DPO (Chen et al., 2025), and RLVR (Wan et al., 2025). We investigate the challenges of applying RLVR in long-context settings and propose a novel framework that substantially enhances its efficacy for alignment.

**Long-Context LLM Agent.** Recent works (Zhao et al., 2024; Qian et al., 2024; Zhang et al., 2024; Zhou et al., 2024) propose utilizing agentic workflows to tackle long-context tasks. Instead of processing the full context via a single LLM pass, these methods split the text into chunks, processing them sequentially and integrating information through multi-turn collaboration, such as updating states in a chain (Zhang et al., 2024). These approaches circumvent the inherent limitations of long-context capabilities in standard LLMs, making them orthogonal to our contribution. Our work focuses on improving the model's native reasoning ability over the full long context. Furthermore, our approach is complementary: it has the potential to enhance agentic frameworks by enabling agents to process larger chunks per step, thereby scaling to even longer contexts.

## 6 CONCLUSION

In this work, we addressed a fundamental limitation of Reinforcement Learning with Verifiable Rewards (RLVR) in long-context scenarios: its inability to effectively learn contextual grounding due to sparse, outcome-only rewards. We formally identified this issue as the "vanishing grounding gradient" problem, where the learning signal for retrieving evidence diminishes significantly with the complexity of the task. To overcome this, we introduced LongRLVR, a novel training paradigm that augments the standard answer reward with a verifiable context reward. This dense reward signal explicitly teaches the model to first identify and extract relevant evidence before generating an answer. Our extensive experiments demonstrate that LongRLVR substantially outperforms both SFT and naive RLVR baselines across multiple models and benchmarks. Our analyses confirm that this success stems from the synergy between the context and answer rewards for both improved grounding and answer quality. By directly training models to ground their reasoning in provided evidence, LongRLVR provides a robust and effective framework for unlocking the long-context reasoning capabilities of LLMs.

## ACKNOWLEDGMENTS

This project was partially supported by the Singapore Ministry of Education Academic Research Fund Tier 1 (Award Number: T1 251RES2514) and MiroMind AI Research Intern Program.

## REPRODUCIBILITY STATEMENT

We have made extensive efforts to ensure the reproducibility of our work. All theoretical claims are formally proven in the appendix, with detailed, step-by-step derivations provided for both RE-INFORCE and GRPO estimators in Appx. §A. The synthetic data generation pipeline, which is crucial for our method, is described in Alg. 1 and further detailed in Appx. §B, covering corpus sourcing, preprocessing, and quality control. All implementation details, including model specifics, training hyperparameters, and the learning strategy, are documented in §3.1. The evaluation protocol, including baselines, benchmarks, and metrics, is clearly outlined in §3.2. To facilitate direct replication of our results, we will release our source code, the generated dataset, and trained model checkpoints upon publication.

## THE USE OF LARGE LANGUAGE MODELS (LLMS)

We utilized Large Language Models (LLMs), including Google's Gemini and OpenAI's GPT series, as assistive tools in the preparation of this manuscript. Their use was limited to the following tasks:

- Generating Python code for the data visualizations in Figures 1, 3, 4, and 5.
- Assisting with the LaTeX formatting of complex elements, particularly Table 1.
- Proofreading and copy-editing the text for grammatical correctness and clarity.

The core research ideation, theoretical contributions, experimental design, and interpretation of results are entirely the work of the human authors. LLMs served strictly as productivity and presentation aids.

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

# A    DETAILED PROOFS FOR PROPOSITIONS 1 AND 2

This appendix provides detailed derivations for the theoretical results presented in Section 2.2 and Section 2.3. We formally prove that outcome-only rewards lead to vanishing gradients for contextual grounding and show how the proposed context reward resolves this issue. The proofs are provided for both the standard REINFORCE policy gradient estimator and the Group-Relative Policy Optimization (GRPO) algorithm.

## A.1    PRELIMINARIES AND NOTATION

We begin by summarizing the formal setup used throughout the proofs. The policy is factorized into a grounding head and an answer head, such that $\pi_\theta(y, Z \mid X, Q) = \pi_\theta^{\text{gnd}}(Z \mid X, Q) \cdot \pi_\theta^{\text{ans}}(y \mid X, Q, Z)$. Our analysis focuses on the gradients with respect to the parameters of the grounding head, $\pi_\theta^{\text{gnd}}$.

**Grounding Head.**    The long context $X$ is partitioned into a set of chunks $C = \{c_1, \ldots, c_N\}$. The grounding head models the selection of each chunk $c_j$ via a binary selection vector $Z = (z_1, \ldots, z_N) \in \{0, 1\}^N$, where $z_j = \mathbf{1}\{c_j \text{ is selected}\}$. We parameterize the grounding policy as a log-linear distribution over subsets

$$\pi_\theta^{\text{gnd}}(Z) = \frac{1}{Z(\theta)} \exp\Big( \sum_{j=1}^{N} s_j z_j + \psi(Z) \Big), \tag{9}$$

where $s_j$ is the logit associated with chunk $c_j$, $\psi(Z)$ is an arbitrary potential that can capture dependencies between chunks, and $Z(\theta)$ is the normalizing constant. This family subsumes the independent Bernoulli model used in the initial version of the paper as the special case $\psi(Z) \equiv 0$. We write $p_j = \mathbb{E}_\theta[z_j] = \Pr_\theta(c_j \in Z)$ for the marginal selection probability. Differentiating $\log \pi_\theta^{\text{gnd}}(Z)$ with respect to $s_j$ yields the score function

$$\nabla_{s_j} \log \pi_\theta^{\text{gnd}}(Z) = z_j - p_j,$$

which is the only property of the policy we use in the subsequent analysis.

**Ground-Truth and Reward.**    Let $G \subseteq C$ be the ground-truth set of essential evidence chunks required to answer the question, with $|G| = g$. We define the "success" event $S$ as the selection of all essential chunks, i.e., $S \equiv \{Z \supseteq G\}$. The probability of this event is $q \triangleq \Pr_\theta(S)$. Under the **Sparse Answer Reward** (Assumption 1), the conditional expected answer reward can be written as $\mathbb{E}[r_{\text{ans}} \mid Z] = \mu_0 + f(Z \cap G)$ for some monotone set function $f : 2^G \to \mathbb{R}_{\geq 0}$ with $f(\emptyset) = 0$. For each $c_j \in G$ and subset $T \subseteq G \setminus \{c_j\}$ we define the marginal gain

$$\Delta_j(T) \triangleq f(T \cup \{c_j\}) - f(T),$$

and assume it is bounded by $\Delta_j(T) \leq \bar{\delta}_j$ for some constant $\bar{\delta}_j > 0$. The all-or-nothing reward used in the initial version corresponds to $f(T) = \delta \cdot \mathbf{1}\{T \supseteq G\}$, where $\Delta_j(T)$ is non-zero only when $T$ already contains all other evidence in $G$.

For the proof of Proposition 2, we additionally use an **Additive Context Reward** of the form

$$r_{\text{ctx}}(Z, G) = \sum_{c_k \in G} \alpha_k z_k, \qquad \alpha_k > 0,$$

so that the total reward is $r_{\text{total}} = r_{\text{ans}} + r_{\text{ctx}}$.

**Policy Gradient Estimators.**    The gradient of an expected reward $\mathbb{E}[R(Z)]$ is computed using the REINFORCE identity (the score function estimator):

$$\nabla_{s_j} \mathbb{E}[R(Z)] = \mathbb{E}\big[ R(Z) \nabla_{s_j} \log \pi_\theta^{\text{gnd}}(Z) \big] = \mathbb{E}\big[ R(Z) (z_j - p_j) \big]. \tag{10}$$

Using a baseline $b$ that does not depend on $z_j$, this is equivalent to the covariance between the reward and the action score:

$$\nabla_{s_j} \mathbb{E}[R(Z)] = \mathbb{E}\big[ (R(Z) - b) (z_j - p_j) \big] = \text{Cov}\big( R(Z), z_j \big). \tag{11}$$

## A.2 PROOF OF PROPOSITION 1: VANISHING GRADIENTS FOR OUTCOME-ONLY REWARDS

**Proposition 1.** *Under Assumption 1, the gradient of the expected answer reward with respect to the logit $s_j$ for any essential chunk $c_j \in G$ satisfies*

$$\nabla_{s_j} \mathbb{E}[r_{\text{ans}}] = \text{Cov}\big(f(Z \cap G), z_j\big) = p_j(1-p_j)\big(\mathbb{E}[f(Z \cap G) \mid z_j{=}1] - \mathbb{E}[f(Z \cap G) \mid z_j{=}0]\big),$$

*and is bounded as*

$$0 \leq \nabla_{s_j} \mathbb{E}[r_{\text{ans}}] \leq p_j(1-p_j)\, \bar{\delta}_j \, \Pr_{\theta}(\mathcal{E}_j),$$

*where $\mathcal{E}_j \triangleq \{Z : \Delta_j((Z \cap G) \setminus \{c_j\}) > 0\}$ is the activation event for $c_j$.*

*Proof using REINFORCE.* Using the covariance form of the policy gradient from Eq. (11), we have

$$\nabla_{s_j} \mathbb{E}[r_{\text{ans}}] = \text{Cov}(r_{\text{ans}}, z_j) = \text{Cov}(\mu_0 + f(Z \cap G), z_j) = \text{Cov}(f(Z \cap G), z_j).$$

For a binary variable $z_j \in \{0, 1\}$, the covariance admits the standard decomposition

$$\text{Cov}(f(Z \cap G), z_j) = p_j(1-p_j)\big(\mathbb{E}[f(Z \cap G) \mid z_j{=}1] - \mathbb{E}[f(Z \cap G) \mid z_j{=}0]\big).$$

To interpret the difference of conditionals, consider the subset of ground-truth chunks other than $c_j$ that are selected, $T(Z) \triangleq (Z \cap G) \setminus \{c_j\} \subseteq G \setminus \{c_j\}$. When $z_j = 1$ we can write

$$f(Z \cap G) = f(T(Z) \cup \{c_j\}) = f\big(T(Z)\big) + \Delta_j\big(T(Z)\big),$$

where $\Delta_j(T)$ is the marginal gain defined above. Taking expectations and subtracting the case $z_j = 0$ yields

$$\mathbb{E}[f(Z \cap G) \mid z_j{=}1] - \mathbb{E}[f(Z \cap G) \mid z_j{=}0] = \mathbb{E}\big[\Delta_j(T(Z)) \mid z_j{=}1\big].$$

By monotonicity, $\Delta_j(T) \geq 0$, and by boundedness, $\Delta_j(T) \leq \bar{\delta}_j$. Let $\mathcal{E}_j = \{Z : \Delta_j(T(Z)) > 0\}$ be the event that $c_j$ has a strictly positive marginal gain given the other selected evidence. We thus obtain

$$0 \leq \mathbb{E}[\Delta_j(T(Z)) \mid z_j{=}1] \leq \bar{\delta}_j \, \Pr_{\theta}(\mathcal{E}_j \mid z_j{=}1) \leq \bar{\delta}_j \, \Pr_{\theta}(\mathcal{E}_j)$$

Substituting back gives the claimed upper bound $\nabla_{s_j} \mathbb{E}[r_{\text{ans}}] \leq p_j(1-p_j)\, \bar{\delta}_j \, \Pr_{\theta}(\mathcal{E}_j)$, and the non-negativity of the gradient follows from the monotonicity of $f$. ∎

*Proof using GRPO.* GRPO uses a group-relative baseline. For a group of $K \geq 2$ i.i.d. trajectories, the unclipped GRPO surrogate gradient at $\theta = \theta_{\text{old}}$ is proportional to the covariance:

$$\nabla_{s_j} \mathcal{L}_{\text{GRPO}}(\theta_{\text{old}}) = \frac{K-1}{K} \text{Cov}(r_{\text{ans}}, z_j) = \frac{K-1}{K} \nabla_{s_j} \mathbb{E}[r_{\text{ans}}].$$

Therefore, the GRPO gradient inherits the same bound from Proposition 1, i.e., it is also scaled by the activation probability $\Pr_{\theta}(\mathcal{E}_j)$ and becomes vanishingly small when $\Pr_{\theta}(\mathcal{E}_j)$ is tiny. ∎

## A.3 PROOF OF PROPOSITION 2: NON-VANISHING GROUNDING SIGNAL

**Proposition 2.** *For a total reward $r_{total} = r_{ans} + r_{ctx}$ where $r_{ctx}(Z, G) = \sum_{c_k \in G} \alpha_k z_k$ with $\alpha_k > 0$, the gradient of the expected total reward with respect to the logit $s_j$ for any essential chunk $c_j \in G$ is*

$$\nabla_{s_j} \mathbb{E}[r_{\text{total}}] = \nabla_{s_j} \mathbb{E}[r_{\text{ans}}] + \alpha_j \text{Var}(z_j) + \sum_{\substack{k \neq j \\ c_k \in G}} \alpha_k \text{Cov}(z_k, z_j).$$

*In particular, combining this with Proposition 1 shows that the answer-only part is at most $p_j(1-p_j)\, \bar{\delta}_j \, \Pr_{\theta}(\mathcal{E}_j)$, while the term $\alpha_j \text{Var}(z_j) = \alpha_j p_j(1-p_j)$ is a dense contribution that does not depend on $\Pr_{\theta}(\mathcal{E}_j)$. If the grounding policy exhibits non-negative correlations among related chunks (so that $\text{Cov}(z_k, z_j) \geq 0$ for $k \neq j$), then*

$$\nabla_{s_j} \mathbb{E}[r_{\text{total}}] \geq \alpha_j \text{Var}(z_j) = \alpha_j p_j(1-p_j) > 0$$

*whenever $p_j \in (0, 1)$.*

*Proof using REINFORCE.* By linearity of expectation, the gradient decomposes: $\nabla_{s_j} \mathbb{E}[r_{\text{total}}] = \text{Cov}(r_{\text{ans}}, z_j) + \text{Cov}(r_{\text{ctx}}, z_j)$. From Proposition 1, we know $\text{Cov}(r_{\text{ans}}, z_j) = \nabla_{s_j} \mathbb{E}[r_{\text{ans}}]$ for $j \in G$. We compute the contribution from the context reward, $r_{\text{ctx}}(Z, G) = \sum_{k \in G} \alpha_k z_k$:

$$\text{Cov}(r_{\text{ctx}}, z_j) = \text{Cov}\Big(\sum_{k \in G} \alpha_k z_k, z_j\Big) = \sum_{k \in G} \alpha_k \text{Cov}(z_k, z_j) = \alpha_j \text{Var}(z_j) + \sum_{\substack{k \neq j \\ c_k \in G}} \alpha_k \text{Cov}(z_k, z_j).$$

Substituting this expression for $\text{Cov}(r_{\text{ctx}}, z_j)$ yields the claimed form for $\nabla_{s_j} \mathbb{E}[r_{\text{total}}]$. The term $\alpha_j \text{Var}(z_j) = \alpha_j p_j (1 - p_j)$ is always non-negative and does not depend on the rare activation event $\mathcal{E}_j$, so it provides a dense per-chunk learning signal even when the answer-only component is nearly zero. When related chunks tend to co-occur, the cross-covariances $\text{Cov}(z_k, z_j)$ further amplify this signal. ∎

**Verification for GRPO and Direct Differentiation.** The GRPO gradient is similarly scaled by $(K-1)/K$, yielding

$$\nabla_{s_j} \mathcal{L}_{\text{GRPO}}(\theta_{\text{old}}) = \frac{K-1}{K}\big(\nabla_{s_j} \mathbb{E}[r_{\text{ans}}] + \text{Cov}(r_{\text{ctx}}, z_j)\big),$$

so the non-vanishing term $\alpha_j \text{Var}(z_j)$ appears unchanged. In the special case where chunk selections are independent and all weights are equal ($\alpha_k \equiv \alpha$), we have $\text{Cov}(z_k, z_j) = 0$ for $k \neq j$ and $\text{Var}(z_j) = p_j(1 - p_j)$, giving

$$\nabla_{s_j} \mathbb{E}[r_{\text{total}}] = \delta \cdot q(1 - p_j) + \alpha \cdot p_j(1 - p_j),$$

which matches the simpler formula reported in the main text of the initial submission. Under the same independence assumptions, direct differentiation of the expected total reward, $\mathbb{E}[r_{\text{total}}] = \mu_0 + \delta q + \alpha \sum_{k \in G} p_k$, also yields the same result.

## B   DATA CURATION AND GENERATION DETAILS

This section provides a comprehensive overview of the pipeline used to generate the grounded long-context question-answering dataset for training LongRLVR.

### B.1   CORPUS SOURCING AND PREPROCESSING

Our data generation process began with a large corpus of long documents from diverse domains, inspired by Gao et al. (2025). Book and arXiv documents were sourced from the Long-Data-Collection dataset, while code documents were sourced from the StarCoder dataset (Li et al., 2023), where all files within a repository were concatenated to form a single document. We filtered this raw corpus to retain only documents with token lengths between 8K and 64K tokens, as measured by the Qwen2.5 tokenizer. This step yielded an intermediate corpus of approximately 18K book, 16K arXiv, and 17K code documents.

### B.2   DOCUMENT SEGMENTATION AND SEMANTIC CLUSTERING

To prepare documents for evidence identification, each document was partitioned into exactly 64 segments. This process was sentence-aware, ensuring splits occurred only at natural text boundaries (e.g., after a period or a newline) to preserve the semantic integrity of each chunk. All segments were then embedded into a high-dimensional vector space using the BGE-M3 sentence encoder (Chen et al., 2024). We applied the DBSCAN algorithm (Ester et al., 1996) to the embeddings within each document, grouping semantically related segments into thematic clusters that would form the basis for targeted question generation.

### B.3   QA GENERATION AND QUALITY CONTROL

We employed a multi-stage generation and filtering process to ensure the final dataset was of high quality. For each document, we randomly selected 4 distinct semantic clusters (with a minimum of 4 chunks each) and prompted the Qwen3-235B-A22B model (Qwen, 2025) to generate 3 candidate

$(Q, y, G)$ tuples per cluster, where $G$ is the set of evidence chunks the model deemed necessary. To maintain high standards, we used the same model as an automated judge to assign a quality rating from 1 to 10 for each generated pair, based on clarity, correctness, and evidence relevance. Both generation and judging used chain-of-thought prompting. A two-stage rejection sampling process then selected the single best QA pair for each document: first, we selected the top-scoring candidate within each cluster, and second, we selected the best among these four candidates. As a final quality filter, we discarded any pair that received a final rating below 9. This pipeline resulted in our final dataset of 46K documents, each paired with a single, high-quality, and well-grounded question-answer pair.

---

**Dataset Example: Long Context QA**

**Question:** What factors contributed to the Mehrikans' eventual disappearance despite their unexpected military victory over the European armada?

**Answer:** Although they achieved a decisive naval victory against the European alliance (a war sparked by their own **commercial greed**), their civilization ultimately collapsed due to **drastic climatic shifts** and **physiological degeneration** (nervous diseases) that reduced their population from 90 million to 12 million.

---

**Context (Excerpt):**
`<CHUNK_0>` THE LAST AMERICAN By J. A. Mitchell ... [Illustration: "–In the soft earth was the imprint of human feet!"] ...
[...]

`<CHUNK_3>` He holds the opinion ... that the Mehrikans were a mongrel race ... wealth, luxury, and gradual decline of the native population; the **frightful climatic changes which swept the country like a mower's scythe**; ... all this is told by Noz-yt-ahl with force and accuracy.

[...]

`<CHUNK_29>` "There were many causes," he answered. ... the effect of climate upon succeeding generations was fatal. **They became flat-chested and thin**, ... **Nervous diseases unknown to us wrought deadly havoc**. ... the **population decreased from ninety millions to less than twelve millions**. ... The **temperature would skip in a single day from burning heat to winter's cold**. No constitution could withstand it...

[...]

`<CHUNK_59>` I have spoken of the Mehrikans being a greedy race. And their greed, at last, resulted in this war. By means of one-sided laws ... **they secured for themselves a lion's share of all profits** ... until at last the leading powers of Europe combined in self-defence...

[...]

`<CHUNK_61>` ... It lasted just one summer afternoon. But the **Mehrikans it was who sent their enemies to the bottom**. And the sea beneath our feet is strewn with iron hulks.

---

**Reference Chunk IDs:** [3, 29, 59, 61]

---

Figure 6: An example instance of training data. The answer requires synthesizing information about the war's outcome (Chunk 61), the war's cause (Chunk 59), and the specific biological and environmental causes of extinction (Chunk 29).

