# OpenReview forum: "LongRLVR: Long-Context Reinforcement Learning Requires Verifiable Context Rewards"
_ICLR.cc/2026/Conference — ICLR 2026 Poster_

### Official Review · Reviewer_RAAP · 2025-10-31

**Soundness:** 3
**Presentation:** 3
**Contribution:** 2
**Rating:** 2
**Confidence:** 4

**Summary:**

This paper focuses on improving long-context reasoning through Rule-based Rewards. They propose a new pipeline to generate data, thereby enabling the use of rule-based rewards to evaluate the evidence used in the LLM output.  Experiments show consistent gains over answer-only baselines, supporting the method’s effectiveness.

**Strengths:**

The problem is significant, as RLVR may stimulate hallucinations and render the training process unstable, while its sparse rewards make effective exploration challenging in practice.

The paper addresses the issue of vanishing gradient in RLVR under sparse outcome-reward settings, examining its causes and implications.

The choice of the F1 score as a reward makes sense to me, since it balances precision and recall rather than encourages the model to cover the evidence as much as possible.

The experiments appear to support the authors’ claims and show consistent improvements.

**Weaknesses:**

However, my concerns arose from the data generation pipeline and the usage of the verifier LLM.

1. It seems that the method is only applicable for the Grounded QA, where evidence can be cleanly chunked. However, in such a case, performing rule-based rewards for the evidence suggestion should be straightforward. The usage of the F1-score is also straightforward to me, since recall encourages the policy to cover as many chunks as possible.
2. A separate verifier LLM is used, which helps identify the evidence and check its alignments with the evidence library; however, it makes the comparison with RLVR unfair. Moreover, an additional LLM can do more (e.g., directly judge whether the final answer matches the reasoning path). Why not use semantic similarity or other similar metrics?
3. The computational and human costs are nontrivial, both from data collection and the additional LLM verifier. Therefore, I am wondering: do the gains adequately justify the substantial supervision and computation?

**Questions:**

See weakness

---

> ### Author Response · Authors · 2025-11-13
> **Quick Clarification on Some Misunderstandings**
>
> Dear Reviewer RAAP,
>
> Thank you for your feedback. We'd like to offer a quick clarification on a few points, as we believe there may be some misunderstandings about our method and experimental setup.
>
> ---
>
> #### **1. On the method's applicability beyond "cleanly chunked" evidence.**
>
> Our goal is to improve general long-context reasoning, and we argue that robust contextual grounding is the key skill required.
>
> The grounded QA setup in our training is a means to an end: it provides a clean, reliable reward signal to teach the model this grounding skill, hence ensuring correct and reliable RL training. The learned ability, however, is not limited to this training format.
>
> This is best shown by our results on **LongBench v2**. That benchmark includes messy, realistic tasks like detective QA, where clues are scattered and hidden, not cleanly chunked. In practice, we saw our `Qwen2.5-7B-LongRLVR` model successfully pull evidence from **over 50 different chunks** to solve a single problem. This shows the model learns a general skill for identifying relevant information across a long and realistic document, which is broadly useful for complex reasoning tasks.
>
> ---
>
> #### **2. On the fairness of using a verifier LLM.**
>
> We took care to ensure the comparison with the naive RLVR baseline was completely fair. The confusion may stem from the two-part nature of our reward system.
>
> 1.  **Answer Reward (`r_ans`):** We use an LLM judge to score the final answer's correctness. It will give more correct and unbiased answer reward to ensure stable and correct RL training. This is a common technique in recent RLVR work (e.g., [1]). **Crucially, the exact same LLM judge was used for both our LongRLVR models and the naive RLVR baselines.** This makes it a controlled, apples-to-apples comparison.
> 2.  **Context Reward (`r_ctx`):** This reward component involves **no LLM at all**. It is a simple, deterministic F-score calculated by comparing the model's selected chunks against the ground-truth annotations in our dataset.
>
> To be clear, **no external LLM verifier is used during inference**. Regarding the semantic similarity, we do use it in our data pipeline to gather relevant chunks. This provides relevance signals in our data,  hence we can use the F-score for the context reward during training.
>
> [1] S3: You Don't Need That Much Data to Train a Search Agent via RL
>
> ---
>
> #### **3. On whether the gains justify the costs.**
>
> We strongly believe they do. The costs are a one-time, automated effort, while the gains are significant and efficient.
>
> First, as mentioned in (2), the "additional LLM verifier" isn't an extra cost of our method; it's a standard part of the RLVR framework that our baseline also uses. The main cost is our data generation, which is a **fully automated pipeline** with no human labeling.
>
> In return for this one-time data synthesis, the performance gains are substantial. For example, on RULER-QA, our method boosts the Qwen2.5-14B model from 73.17 to **88.90 (+15.7 points)**, producing large performance improvements.
>
> This approach is also highly efficient. Our LongRLVR-trained **14B** model outperforms the much larger Qwen2.5-**72B**-YaRN model (46.5 vs 43.5 on LongBench v2), while comparable to QwenLong-L1-**32B** which also involves long-context RLVR. Achieving superior performance with a significantly smaller size demonstrates the power and computational efficiency of our training strategy.
>
> Ultimately, our work identifies the context reward as the missing piece for making RLVR work in long-context scenarios. We believe this core insight will be valuable for the community. We provide one effective solution as LongRLVR, that said, it has potential to be followed and improved with more efficient strategies in the future.

---

> > ### Author Response · Authors · 2025-11-20
> > **Request for feedback on clarifications regarding method fairness and generalizability**
> >
> > Dear Reviewer RAAP,
> >
> > We are writing to kindly follow up on our previous response regarding your concerns about the **data generation pipeline** and the **verifier LLM**.
> >
> > We believe there may be some **fundamental misunderstandings** regarding the experimental setup that influenced your assessment. Specifically, our rebuttal clarifies that:
> > 1.  **Fairness:** The LLM judge is a standard component used identically in both our method and the baseline, ensuring a strictly fair comparison.
> > 2.  **Applicability:** The method teaches general grounding skills that extend beyond "cleanly chunked" data, as evidenced by our strong performance on the complex LongBench v2.
> > 3.  **Cost:** The pipeline is fully automated and the efficiency gains (e.g., our 14B model outperforming a 72B model) justify the one-time data synthesis.
> >
> > We would appreciate it if you could review our clarifications. We **remain fully open to addressing any further concerns or questions** you might have once these points are cleared up.
> >
> > Best regards,
> >
> > The Authors

---

> > > ### Comment · Reviewer_RAAP · 2025-11-27
> > >
> > > Thanks for the authors’ response clarifying the use of rewards in the baseline methods. I have raised my score, but I still believe the proposed approach is quite costly in terms of collecting evidence and constructing evidence blocks. Moreover, collecting more evidence does not necessarily mean that the LLM can reliably answer the question, and vice versa.

---

> > > > ### Author Response · Authors · 2025-12-03
> > > > **Reply to remaining concerns on Data Cost and Method Applicability**
> > > >
> > > > Dear Reviewer RAAP,
> > > >
> > > > Thank you for your response and for raising your score. We appreciate the opportunity to clarify these two remaining points regarding the cost and the relationship between evidence collection and answer reliability.
> > > >
> > > > **1. On the cost of collecting evidence and constructing blocks.**
> > > > We would like to clarify that the cost of "collecting evidence and constructing blocks" is marginal regarding data collection, training, and inference.
> > > >
> > > > *   **Efficient Data Generation:** We do not use LLMs to scan vast amounts of text to "find" evidence for existing queries. Instead, we use a cost-effective **reverse-generation** pipeline:
> > > >     1.  We use a standard clustering algorithm on embeddings from **BGE-M3** (an extremely small 560M parameter model).
> > > >     2.  We then prompt the generator LLM (Qwen3-235B) to synthesize a QA pair *based on* a specific cluster.
> > > >     This means the evidence is natively grounded without expensive search or human annotation.
> > > > *   **Reusable Asset:** The computational cost is a one-time investment for data synthesis. As our dataset consists of high-quality, long-context QA pairs with grounded evidence, it is not narrow to our specific paper but serves as a general-purpose community asset, further justifying the initial compute.
> > > > *   **Negligible Inference Cost:** During training and inference, the only additional cost for collecting evidence is the generation of chunk IDs (a few tokens), which is negligible compared to the computational cost of generating the full long-context response.
> > > >
> > > >
> > > > ----
> > > >
> > > > **2. On the concern that "Collecting more evidence does not mean reliable answers" and applicability.**
> > > > We agree that simply selecting *more* evidence does not guarantee a correct answer. However, we would like to clarify that our method is designed to ensure the *correct* evidence is selected to enable reliable reasoning, even when that evidence is not "cleanly chunked."
> > > >
> > > > Our method is not limited to tasks with pre-defined or "clean" evidence blocks on both training and inference.
> > > >     *   **Training Data:** Our training sources are natively long raw texts (books, papers, and code repositories) where evidence is implicitly hidden and scattered, not cleanly pre-chunked. The "blocks" are merely a mechanism for the reward model, not a constraint on the data source.
> > > >     *   **Evaluation:** Our strong performance on benchmarks like **LongBench v2**—which contains realistic, information-intensive queries with scattered clues—demonstrates that our model learns to effectively scan and select necessary (**even implicit with large amount**) evidence from complex, unstructured contexts.
> > > >
> > > > As to the answer reliability, the large gains of our method over naive RLVR on these challenging benchmarks **have already demonstrate the effectiveness.**
> > > >
> > > >
> > > > Best regards,
> > > >
> > > > The Authors

---

> ### Public Comment · ~Xin_Men1 · 2025-11-13
> **request for opensource dataset**
>
> Do you have any plan to opensource you training data and Qwen2.5-7B/14B-LongRLVR?

---

> > ### Author Response · Authors · 2025-11-13
> >
> > Thank you for your interest! We are actively in the process of releasing the code, data, and models. It will be publicly available very soon, and in accordance with the anonymity policy.

---

### Official Review · Reviewer_fzK1 · 2025-11-01

**Soundness:** 3
**Presentation:** 3
**Contribution:** 2
**Rating:** 4
**Confidence:** 4

**Summary:**

This paper addresses a fundamental limitation of RLVR when applied to long-context reasoning tasks. The authors identify that outcome-only rewards suffer from vanishing gradients for the contextual grounding process. They formally prove this vanishing gradient problem and propose LongRLVR, which augments sparse answer rewards with dense, verifiable context rewards that explicitly supervise evidence selection. The method is validated on RULER-QA, LongBench v2, and LongReason benchmarks, showing consistent improvement against vanilla GRPO and SFT. The approach requires ground-truth evidence annotations, and the authors also propose a data generation pipeline using clustering and rejection sampling.

**Strengths:**

1. The formal analysis of why the outcome-only reward is insufficient for the long-context retrieval-based task provides some transferable insights.
2. The modulated F-score reward, combining unconditional grounding reward and synergistic success reward, is thoughtfully designed.
3. The paper provides extensive analysis on both synthetic and real-world long-context tasks, and the paper includes thorough ablations examining reward components, data quality, hyperparameters, and chunk number robustness, making the claims more sound.

**Weaknesses:**

1. The comparison is a bit weak, which hinders the overall soundness of the work. Interleaving reasoning and retrieval is now becoming more popular. I would suggest comparing with some RAG baselines (which do not require RLVR but fit the same scenario), as well as some recent works like [1].
2. Assumption 1 seems too strong for the analysis. In reality, the reward for retrieved evidence, if applied, should be more continuous than the 0 or 1 sparse reward. Also, the independence assumption for chunk selection might not be true since the evidence should be highly related in a multi-hop QA scenario like the ones in LongBench.
3. A human evaluation of the validity of the generated data or some examples provided would be very beneficial.
4. There is a potentially biased evaluation regarding the training data. The vanilla RLVR and SFT baselines are trained on the same generated data with evidence, but they don't have the corresponding training signal, which may introduce bias towards the proposed method.

[1] Wang et al. 2025. Improving Context Fidelity via Native Retrieval-Augmented Reasoning. arXiv:2509.13683.

**Questions:**

1. Would it be helpful if some existing QA datasets with ground-truth evidence (like HotpotQA) is used partially as the training data or as a seed dataset for the data generation process?
2. Are the chunk identifiers ([CHUNK_N]) added as new special tokens?
3. The useful chunks are generated after the thinking process. Does the model actually use or refer to the chunks during the training process?
4. How does performance degrade with noisy evidence annotations?
5. Can you evaluate on datasets with evidence annotations (e.g., HotpotQA) for the retrieval accuracy?

---

> ### Author Response · Authors · 2025-11-20
> **Rebuttal to Reviewer fzK1 (1/N)**
>
> Dear Reviewer fzK1,
>
> We sincerely thank for the thoughtful feedback. We have conducted extensive analyses and additional experiments (*consuming over 4,000 A100 GPU hours*) to address your concerns regarding baselines, theoretical assumptions, and data bias. Below, we address each point in detail. We hope these new results and clarifications convincingly address the raised concerns.
>
> ---
> ### **1. Comparison with RAG Baselines and Concurrent Work (CARE)**
>
>
> We have added comparisons against a standard RAG baseline and the concurrent work CARE.
> *   **RAG Setup:** We use BGE-M3 as the embedding model with a 512-token chunk size, retrieving the top-16 chunks.
> *   **CARE:** We direcly take the officially released checkpoint of CARE based on LLaMA-3.1-8B.
> *   **Evaluation Metric:** Since RL-trained models (like CARE and LongRLVR) often output formats that do not perfectly match the strict regex requirements of LongBench v2, we utilized `Qwen3-235B-A22B-Instruct` as a judge to evaluate the correctness of the answers for *all* models (RAG, CARE, and LongRLVR) to ensure a fair comparison.
>
> **Results on LongBench v2 (LLaMA-3.1-8B backbone):**
>
> | Model                 |  Short   |  Medium  |   Long   | Overall  |
> | :-------------------- | :------: | :------: | :------: | :------: |
> | LLaMA-3.1-8B          |   34.4   |   31.6   |   21.3   |   30.4   |
> | + RAG (BGE-M3)        |   37.2   |   24.2   |   27.8   |   29.6   |
> | + CARE                |   32.2   |   27.9   |   33.3   |   30.6   |
> | + **LongRLVR (Ours)** | **41.1** | **30.7** | **38.9** | **36.2** |
>
> Our LongRLVR significantly performs **better than all baselines** on the challenging LongBench v2.  We found naive RAG cannot work well on the challenging long-context benchmark, falling behind the original long-context baselines. As our method introduces significant gains over original model by RL training, we **exclude the training-free baselines** in our paper.
>
> Note that CARE also involves RL training but on a shorter length. To ensure fairness, we also evaluated LongRLVR on the benchmarks used in the CARE paper (MFQA, HotpotQA, 2WikiMQA, MuSiQue).  CARE reports F1 scores by words matching between the generated answer and ground truths, which we found extremely unstable across different runs. Here we used `Qwen3-235B-A22B-Instruct` as judge to evaluate answer correctness (0 or 1) for both methods to ensure robust comparison.
>
> | Model | MFQA | HotpotQA | 2WikiMQA | MuSiQue | **AVG** |
> | :--- | :---: | :---: | :---: | :---: | :---: |
> | LLaMA-3.1-8B (Base) | 72.7 | 70.5 | 54.5 | 32.5 | 57.5 |
> | CARE | 79.3 | 78.5 | **84.5** | **53.5** | 74.0 |
> | **LongRLVR (Ours)** | **88.0** | **80.0** | 78.0 | 51.0 | **74.3** |
>
> **Conclusion:** LongRLVR achieves performance comparable to or better than CARE. It is worth noting that **CARE is a concurrent work that also adopts a form of "context reward"** using retrieval tags (as we discuss in related work). The success of both methods validates the core hypothesis that outcome-only rewards are insufficient, while LongRLVR provides a more rigorous theoretical grounding to point out where the root benefits come from.
>
> ---
> ### **2. Validity of Assumption 1 (Sparse Rewards & Independence)**
>
>
> We agree that real-world rewards are continuous and evidence chunks are often dependent. Therefore, we have extended our theoretical analysis in the method and appendix (updated in the PDF) to show that **our conclusions hold even under these relaxed conditions.**
>
> Even with a continuous reward function, the "Vanishing Gradient" problem persists, as the reward for a specific chunk depends on the *joint* selection of other prerequisite chunks. This is because, **the optimization gradient for selecting a specific chunk $c_i$ is bounded by the probability that *all other prerequisite evidence* (that makes $c_i$ useful) has already been selected.**
>
> In challenging long-context tasks, the probability of accidentally selecting *all* prerequisites in the initial policy is extremely low. Therefore, even **if the reward is continuous, the gradient for the crucial "missing link" evidence remains near zero**. Our context reward solves this by providing dense feedback for individual chunks, regardless of whether the full chain is complete.

---

> > ### Author Response · Authors · 2025-11-20
> > **Rebuttal to Reviewer fzK1 (2/N)**
> >
> > ### **3. Data Quality and Examples**
> >
> >  We have included a detailed example of our training data in **Figure 6 of the Appendix**. Note that we utilized the state-of-the-art `Qwen3-235B-A22B-Instruct` (in thinking mode) for data generation, which is of high quality. Manual inspection of random samples confirms that the generated questions require true long-context retrieval, synthesis, and reasoning, rather than simple lookups. The example in Figure 6 demonstrates a challenging query requiring the model to locate scattered evidence regarding a war's outcome, its cause, and environmental factors, and then **synthesize** them into a coherent answer.
> >
> >
> > ---
> >
> > ### **4. Potential Bias in Baselines (Access to Evidence Signal)**
> >
> > As the evidence signal in LongRLVR is does what I seek to improve for RLVR, we feel the performance gains are enough and fair to demonstrate the effectiveness of the context reward design. To further address your concern, we conducted a new experiment where we explicitly provided the ground-truth evidence signal to the baselines.
> > 1.  **SFT+:** We added the ground-truth chunk IDs to the loss calculation (supervising the model to predict them).
> > 2.  **RLVR+:** We appended the ground-truth chunk IDs to the prompt suffix, effectively "leaking" the evidence to the model during rollout to see if it could utilize it.
> >
> > **Results on LongBench v2 (Qwen2.5-7B):**
> >
> > | Model | Short | Medium | Long | Overall |
> > | :--- | :---: | :---: | :---: | :---: |
> > | **SFT Baseline** | 36.7 | 32.6 | 28.7 | 33.2 |
> > | SFT + Evidence Signal | 42.8 | 30.7 | 28.7 | 34.6 |
> > | **RLVR (Naive)** | 37.2 | 29.3 | 30.6 | 32.4 |
> > | RLVR + Evidence Signal | 38.3 | 30.7 | 24.1 | 32.0 |
> > | **LongRLVR (Ours)** | **45.6** | **35.8** | **32.4** | **38.6** |
> >
> > **Conclusion:**
> > *   Providing evidence signals to SFT yields minor gains (33.2 $\to$ 34.6) but still falls short of LongRLVR (38.6).
> > *   Providing evidence to naive RLVR does not help (likely because this skips the optimization for contextual grounding.).
> > *   This confirms that our LongRLVR provides a **more effective utilization of the "evidence signal" to improve the long-context capability**, highlighting the effectiveness of proposed method.

---

> > > ### Author Response · Authors · 2025-11-20
> > > **Rebuttal to Reviewer fzK1 (3/N)**
> > >
> > > **Q1: Would using existing datasets (like HotpotQA) as seed/training data help?**
> > >
> > > **A:** We experimented with adding 5k samples from HotpotQA to our training set following the same training settings (train for one epoch). The results are listed below.
> > >
> > > **LongBench v2:**
> > >
> > > | Model                        | Short | Medium | Long | Overall |
> > > |------------------------------|:-----:|:------:|:----:|:-------:|
> > > | Qwen2.5-7B-LongRLVR          | 45.6  | 35.8   | 32.4 | 38.6    |
> > > | + 5K HotpotQA                | 41.1  | 34.9   | 30.6 | 36.2    |
> > >
> > > **RULER-QA:**
> > >
> > > | Model               | 32K  | 64K  | 128K |  AVG  |
> > > | ------------------- | :--: | :--: | :--: | :---: |
> > > | Qwen2.5-7B-LongRLVR | 85.5 | 76.5 | 79.0 | 80.33 |
> > > | + 5K HotpotQA       | 86.4 | 78.6 | 72.5 | 79.17 |
> > >
> > >
> > > We found the performance slightly degraded (e.g., RULER-QA Avg dropped from 80.33 to 79.17) when using HotpotQA training data. This may be because HotpotQA contexts are generally short (<10k tokens), making the contextual grounding not challenging for current models. However, we believe using HotpotQA queries augmented with much larger web-document contexts could be a valuable source to provide high-quality training data in future work.
> > >
> > > ---
> > >
> > > **Q2: Are the chunk identifiers (`[CHUNK_N]`) added as new special tokens?**
> > >
> > > **A:** No. We require the LLM to generate the identifiers as standard text. If `[CHUNK_N]` is added as special token, it will be assigned a randomly initialized embedding, making the LLM unable to correctly generate it. Special tokens are usually added before pretraining (e.g., end token) and just for format token without the requirement of generation (e.g., \<user\>).
> > >
> > >
> > > ---
> > >
> > > **Q3: Does the model actually refer to the chunks during the thinking process?**
> > >
> > > **A:** Yes. We analyzed 10 random rollouts from `Qwen2.5-7B-LongRLVR`. In total, the model outputted 62 chunk IDs in the final `<useful_chunks>` section. Of these, **58 (93.5%)** were explicitly referenced or discussed during the chain-of-thought (thinking) process (some miss items may be out of some reference pattern like \<CHUNK_4\>-\<CHUNK_8\> as we found). We believe current LLMs have enough capability to ensure the appearance of generated chunk ids in thinking process.
> > >
> > > ---
> > > **Q4: How does performance degrade with noisy evidence annotations?**
> > > **A:** We performed two controlled noise experiments for Qwen2.5-7B-LongRLVR:
> > >
> > > 1. **Additive noise:** for each instance, we **randomly add 20% extra chunk IDs** as (false) positives.
> > > 2. **Missing evidence:** for each instance, we **randomly remove 20% of the true evidence chunk IDs**.
> > >
> > > Training and evaluation settings are kept identical otherwise. Results on LongBench v2:
> > >
> > > | Model / Noise Setting                | Short | Medium | Long | Overall |
> > > |--------------------------------------|:-----:|:------:|:----:|:-------:|
> > > | Qwen2.5-7B-LongRLVR (clean)          | 45.6  | 35.8   | 32.4 | 38.6    |
> > > | + 20% added noisy evidence           | 44.4  | 33.5   | 34.3 | 37.6    |
> > > | + 20% removed (missing) evidence     | 41.7  | 31.6   | 27.8 | 34.4    |
> > >
> > > **Observations.**
> > >
> > > - LongRLVR is **reasonably robust** to *added* noisy evidence: performance drops only by about 1 point overall.
> > > - In contrast, **missing evidence annotations** cause a larger degradation. Intuitively, adding extra false-positive chunks primarily reduces **precision**, but **correctly recalled chunks are still positively rewarded**, so the F-score-based context reward remains informative. However, removing true evidence means that rollouts which correctly retrieve these chunks are **mistakenly treated as *incorrect*** by the context reward, causing useful trajectories to receive low advantages.
> > >
> > >
> > > **Q5: Evaluate retrieval accuracy on datasets like HotpotQA.**
> > > **A:** Yes. We evaluated retrieval accuracy on HotpotQA distractor validation set. Each passage in the HotpotQA distractor setting is treated as a **chunk**. For each question, we compare the set of predicted chunks in `<useful_chunks>` with the ground-truth supporting passages. We report:
> > >   - **Precision (Accuracy):** fraction of predicted chunks that are ground-truth evidence.
> > >   - **Recall:** fraction of ground-truth evidence chunks that are retrieved.
> > >
> > > The results are:
> > >
> > > | Model                     | Precision (Acc) | Recall  |
> > > |---------------------------|:--------------:|:-------:|
> > > | LLaMA-3.1-8B              | 62.02          | 60.70   |
> > > | + LongRLVR                | **84.93**      | **95.30** |
> > > | Qwen2.5-7B-1M             | 74.45          | 73.25   |
> > > | + LongRLVR                | **90.24**      | **94.10** |
> > > | Qwen2.5-14B-1M            | 89.10          | 87.90   |
> > > | + LongRLVR                | **91.63**      | **97.50** |
> > >
> > > LongRLVR **substantially improves both precision and recall** across all base models, often pushing recall above 95%. This directly demonstrates that our training not only improves final QA accuracy, but also **significantly enhances the model’s contextual grounding ability**.

---

> > > > ### Author Response · Authors · 2025-11-27
> > > > **Inquiry regarding feedback on our rebuttal**
> > > >
> > > > Dear Reviewer fzK1,
> > > >
> > > > We submitted our response to your review 7 days ago and would like to gently ensure it has reached you. We are very open to further discussion and would appreciate knowing if our response has sufficiently addressed your concerns. We look forward to hearing from you.
> > > >
> > > > Best regards,
> > > > The Authors

---

### Official Review · Reviewer_kmU8 · 2025-11-01

**Soundness:** 4
**Presentation:** 4
**Contribution:** 3
**Rating:** 8
**Confidence:** 2

**Summary:**

This paper proposes LongRLVR, a reinforcement learning framework for long-context LLMs that introduces verifiable context rewards to overcome vanishing gradients in grounding. Instead of rewarding only final answers, LongRLVR adds dense rewards for correctly selecting evidence chunks, ensuring effective credit assignment across lengthy inputs. It decomposes the policy into grounding and answering heads, uses F-score–based context rewards, and achieves large gains on long-context QA benchmarks, outperforming outcome-only RL baselines and enabling smaller models to surpass larger ones.

**Strengths:**

1. **Timely and impactful problem.**
The paper tackles a highly relevant and increasingly important issue — how to perform reinforcement learning effectively in long-context settings. As large-context reasoning becomes central to emerging LLM-based agents and search systems, addressing the credit-assignment and gradient-vanishing challenges identified here is both timely and of broad significance.
2. **Strong motivation, clear formulation, and well-executed methodology.**
The study is well motivated and rigorously executed. It formally defines the reward-vanishing problem in long-context RL, provides theoretical analysis to explain why outcome-only rewards fail, and introduces a principled solution through verifiable context rewards. The inclusion of a synthetic yet well-controlled dataset allows precise testing, and the resulting performance gains over baselines are substantial and convincing.
3. **Clear structure and presentation.**
The paper is clearly organized and well written, with intuitive explanations and consistent notation. The conceptual flow, from identifying the issue to formal analysis, method design, and empirical validation, is easy to follow, making the technical contributions accessible and well supported.

**Weaknesses:**

1. **Strong theoretical assumptions.**
The analysis relies on several simplifying assumptions that may not fully hold in practice. In particular, it adopts an all-or-nothing reward assumption, where the answer reward increases only when the entire evidence set G is selected. In reality, LLMs often produce correct answers from partial or alternative evidence, making this assumption less realistic. Similarly, the independent Bernoulli selection assumption overlooks dependencies between evidence chunks—real policies typically select evidence in a correlated or sequential manner, which could alter the theoretical gradient behavior. It would strengthen the paper to discuss under what scenarios these assumptions are likely to hold (e.g., explicit fact-retrieval tasks) and where they may fail (e.g., open-domain reasoning). Such clarification would help readers understand the practical scope of the theoretical results.
2. **Connection to broader long-context RL not explored.**
This is more like a suggestion. The paper could better relate its formulation and method to other long-context settings, such as agent RL, where the context includes environment state and action history (like adding a discussion section to appendix). Discussing which aspects of LongRLVR’s framework may transfer and which may not would improve generality and impact.

**Questions:**

Some other gradient vanish work in LLM RL could be discussed in related work. e.g, "Vanishing Gradients in Reinforcement Finetuning of Language Models"

---

> ### Author Response · Authors · 2025-11-20
> **Rebuttal to Reviewer kmU8**
>
> Dear Reviewer kmU8,
>
> We sincerely thank for the thoughtful feedback and postive rating. To address your questions, we have updated the paper to improve the theoratical framework and related work discussion. Please find the details below.
>
> ----
> ### **Validity of Assumption 1 (Sparse Rewards & Independence)**
>
>
> We agree that real-world rewards are continuous and evidence chunks are often dependent. Therefore, we have extended our theoretical analysis in the method and appendix (updated in the PDF) to show that **our conclusions hold even under these relaxed conditions.**
>
> Even with a continuous reward function, the "Vanishing Gradient" problem persists, as the reward for a specific chunk depends on the *joint* selection of other prerequisite chunks. This is because, **the optimization gradient for selecting a specific chunk $c_i$ is bounded by the probability that *all other prerequisite evidence* (that makes $c_i$ useful) has already been selected.**
>
> In challenging long-context tasks, the probability of accidentally selecting *all* prerequisites in the initial policy is extremely low. Therefore, even **if the reward is continuous, the gradient for the crucial "missing link" evidence remains near zero**. Our context reward solves this by providing dense feedback for individual chunks, regardless of whether the full chain is complete.
>
> ---
> ### **Connection to broader long-context RL not explored.**
>
> Thank you for this valuable suggestion. We agree that LongRLVR has significant implications for agentic workflows. We have added a discussion in the **Related Work** section to explicitly connect our framework to Agent RL. Standard long-context agents often circumvent the context window limit by splitting history into chunks or using external memory tools. In contrast, our work focuses on improving the model's native capability to reason over the full context end-to-end. Our approach **is complementary to agentic frameworks**. By enhancing the base model's ability to process and ground information within its active context window, LongRLVR can enable agents to handle larger "working memories" and process more complex observations per step, thereby scaling agent capabilities to even longer horizons.
>
> ---
>
> ### **Related work to vanish gradient**
>
> Thank you for the suggestion. As the extremely upper bound of context reward will contribute to a small std, the gradient should vanish as proposed by this work (Vanishing Gradients in Reinforcement Finetuning of Language Models). We now include the discussion in our method section.
>
> ---
>
>
> We hope these clarifications and revisions strengthen the paper and reinforce your positive assessment. Thank you again for your support of our work.
>
> Sincerely

---

> > ### Comment · Reviewer_kmU8 · 2025-11-28
> >
> > Thanks for adding the new theoretical proof. It looks solid to me overall. One caveat is that the current bound only shows that the gradient scales with \Pr(E_j), which is argued to be small but is not formally shown to be exponentially small without the stronger assumptions used before (e.g., independence). Because of that, I think the “exponentially vanishing gradient” claim in the paper should be softened, or made explicitly conditional on those stronger assumptions.

---

> > > ### Author Response · Authors · 2025-12-03
> > >
> > > Dear Reviewer kmU8,
> > >
> > > Thank you for your valuable feedback and for validating the soundness of our updated theoretical proof. We fully agree with your observation: while the gradient scales with the activation probability $\Pr(E_j)$—**which is still vanishingly small** in challenging long-context scenarios—the decay is not strictly 'exponential' without assuming independence among chunk selections.
> > >
> > > Accordingly, we have revised the manuscript to soften these claims. We have removed the unconditional 'exponentially' qualifier from the Abstract, Introduction, Method, and Conclusion sections, instead describing the issue as a 'vanishing gradient' problem driven by the sparsity of the activation event.
> > >
> > > Best Regards,
> > >
> > > The Authors

---

### Meta-Review · Area_Chair_yKzL · 2026-01-08

**Summary:**

This paper introduces LongRLVR, a method to boost long-context reasoning in LLMs by adding dense, verifiable context rewards to standard RLVR, tackling vanishing gradients from sparse outcome rewards. Key contribution: formal proof of the issue plus empirical gains on benchmarks like RULER-QA and LongBench. Reviewers liked the timely problem, solid theory, and results, but noted weak assumptions, missing baselines, and high data costs.

**Reviewer Concerns:**

Rebuttal addressed theory assumptions via generalizations, baselines with new RAG/CARE experiments showing strong outperformance, and data quality with examples and noise tests.


minor worries on cost justification and applicability beyond QA, but these seem secondary given gains.

**Reviewer Scores:**

Reviewer kmU8: likely stays at 8, as they validated updates. Reviewer fzK1: could rise to 6, given extensive new results, though no direct response. Reviewer RAAP: probably up to 5, after clarifications and score bump.

---

### Decision · Program_Chairs · 2026-01-26

Accept (Poster)